# Genome-Wide Identification and Analysis of the *MYB* Transcription Factor Gene Family in Chili Pepper (*Capsicum* spp.)

**DOI:** 10.3390/ijms22052229

**Published:** 2021-02-24

**Authors:** Magda L. Arce-Rodríguez, Octavio Martínez, Neftalí Ochoa-Alejo

**Affiliations:** 1Departamento de Ingeniería Genética, Unidad Irapuato, Centro de Investigación y de Estudios Avanzados del Instituto, Politécnico Nacional, 36824 Irapuato, Guanajuato, Mexico; mlarcerodriguez@ucdavis.edu; 2Unidad de Genómica Avanzada (Langebio), Centro de Investigación y de Estudios Avanzados del Instituto Politécnico Nacional (Cinvestav), 36824 Irapuato, Guanajuato, Mexico

**Keywords:** *Capsicum*, chili pepper, MYB transcription factor, expression profiles, secondary metabolism, capsaicinoids, carotenoids, phenylpropanoids, lignins, vitamin C

## Abstract

The MYB transcription factor family is very large and functionally diverse in plants, however, only a few members of this family have been reported and characterized in chili pepper (*Capsicum* spp.). In the present study, we performed genome-wide analyses of the MYB family in *Capsicum annuum*, including phylogenetic relationships, conserved domain, gene structure organization, motif protein arrangement, chromosome distribution, chemical properties predictions, RNA-seq expression, and RT-qPCR expression assays. A total of 235 non-redundant MYB proteins were identified from *C. annuum*, including R2R3-MYB, 3R-MYB, atypical MYB, and MYB-related subclasses. The sequence analysis of CaMYBs compared with other plant MYB proteins revealed gene conservation, but also potential specialized genes. Tissue-specific expression profiles showed that *CaMYB* genes were differentially expressed, suggesting that they are functionally divergent. Furthermore, the integration of our data allowed us to propose strong *CaMYBs* candidates to be regulating phenylpropanoid, lignin, capsaicinoid, carotenoid, and vitamin C biosynthesis, providing new insights into the role of MYB transcription factors in secondary metabolism. This study adds valuable knowledge about the functions of *CaMYB* genes in various processes in the *Capsicum* genus.

## 1. Introduction

Transcription factors (TFs) are sequence specific DNA binding proteins that recognize specific *cis*-elements in the promoters of target genes to activate or repress their expression in response to endogenous or exogenous stimuli and in order to control biochemical and physiological processes. TFs can be divided into many multigene families according to their DNA-binding domains [1]. MYB transcription factors are one of the largest TF families, and they are present in all eukaryotes. The oncogene *v-MYB* was the first *MYB* transcription factor identified in avian myeloblastosis virus [2]. MYB proteins are widely distributed in plants and they have been implicated in the regulation of many biological processes, such as primary and secondary metabolism, plant growth and development, circadian clock control, cell cycling development, response to biotic and abiotic stresses, and plant defense [3]. The first plant *MYB* gene identified was *COLORED1* (*C1*) from *Zea mays*, involved in the regulation of anthocyanin biosynthesis in the aleurone and scutellum tissues of the kernel [4].

MYB TFs are characterized by containing a MYB DNA-binding conserved domain, which is approximately 50–55 amino acid residues in length, with three spaced tryptophan residues, forming a helix-turn-helix (HTH) fold. Conversely, the amino acid sequence outside the conserved MYB domain is highly divergent and responsible for the diverse regulatory activity of the MYB proteins. MYB TFs are classified into four classes depending on the number of MYB repeats: 1R-MYB, R2R3-MYB, 3R-MYB, and 4R-MYB, containing one, two, three, or four imperfect MYB repeats. The 1R-MYBs are also referred to as MYB-related, which typically, but not always, contain a single or partial MYB repeat. MYB-related proteins comprise a collection of several protein subgroups, including CIRCADIAN CLOCK ASSOCIATED1 (CCA-1)-like, I-box-binding factor-like, DIVARICATA (DIV)-like, Telomeric DNA-binding protein (TBP)-like, CAPRICE (CPC)-like, Late Elongated Hypocotyl (LHY)-like, KANADI (KAN)-like, TRYPTYCHON (TRY)-LIKE, Enhancer of TRY and CPC (ETC)-like, Phosphate Starvation Response (PHR)-like, RADIALIS (RAD)-like, Golden1/2 (GLK)-like, Early Flowering MYB (EFM)-like, HRS1 Homolog 5 (HHO5)-like, MYBC1-like, MYR1/2-like, KUODA1 (KUA1)-like, Early Phytochrome Responsive1 (RVE)-like, and Salt-Related MYB1 (SRM1)-like [5,6,7,8,9,10,11,12,13,14]. MYB family TFs have been identified in numerous plant species, with 100–250 members, for example, in Chinese pear 129 PbMYBs were identified [15], rice genome contained 233 OsMYBs [16], Arabidopsis genome included 198 AtMYBs [17], in pineapple, a total of 184 AcMYBs were identified [18], in the tomato genome, 127 SlMYBs were found [19], 245 HaMYBs were identified in sunflower [20], and 253 StMYBs were reported in potato [21].

Chili pepper fruit is a major vegetable and spice crop worldwide, being *Capsicum annuum* the most widely grown in the world [22]. The genome size of *C. annuum* is 3.48 Gb [23,24], in which 2153 TFs were identified (6.25% of the total genes) spanning 80 gene families [23]. A few *CaMYB* genes have been shown to play important roles in the regulation of plant development and secondary metabolism in the *Capsicum* genus. For instance, *CaA* an *R2R3-MYB* gene is involved in the control of anthocyanin biosynthesis [25,26]. *CaMYB31*, *CaMYB108,* and *CaMYB48* have been shown to participate in the regulation of capsaicinoid biosynthesis [27,28,29], and additionally *CaMYB108* is involved in stamen development [28]. *CaBLIND* was found to regulate axillary meristem initiation and flowering transition [30]. Recently, 108 members of the R2R3-MYB gene subfamily were identified in chili pepper, of which six genes were proposed as candidates to be regulating the synthesis of capsaicinoids [31].

The present study was focused on the genome-wide analysis of the MYB family in *C*. *annuum* and a comparison to other known MYB plant proteins, including database searches, phylogenetic relationships, gene structure organization, chromosome distribution, conserved domain analysis, and motif protein composition. Furthermore, the co-expression analysis during chili pepper fruit development and their validation by the RT-qPCR analysis identified *CaMYB* genes that may play important roles in the regulation of phenylpropanoid, lignin, capsaicinoid, carotenoid, and vitamin C biosynthetic pathways. This work provided deep insights into the function of *CaMYBs*, allowing the identification of gene resources for breeding and engineering of *Capsicum* spp. and possibly for other plant species.

## 2. Results

### 2.1. Identification of MYB Family Genes in Capsicum annuum

The identification of *MYB* family genes in *C. annuum* genome (cultivar Zunla-1) was carried out with a Blastp search using as query the Pfam domains: PF00249 and PF13921. We identified 345 proteins with at least one MYB domain, which were localized in 235 loci, representing almost 11% of TFs in *Capsicum*. The presence of the MYB domain was verified using the Pfam database [32]. In total, 116 R2R3-MYBs, five 3R-MYBs, 92 MYBs-related, and two atypical MYB (CaMYB5R and CaMYBCDC) were identified in *C. annuum*. Interestingly, we did not identify any 4R-MYB, but a single gene with five MYB repeats was detected. In addition, the domain analysis showed the presence of proteins with other conserved family domains alongside the MYB-repeat, including: nine response regulator receiver domain, five linker-histone, two zinc finger zz type, and four SWIRM domains. The identified *CaMYB* genes ranged from 237 to 3465 pb, and the corresponding proteins ranged from 78 to 1154 amino acids in length. The protein mass ranged from 8.96 to 183.49 kDa, while the isoelectric point (pI) values ranged from 4.34 to 9.36. Additionally, we predicted the subcellular localization of CaMYB proteins, of which 90.2% presented a unique Nuclear Localization Signal (NLS), 7.65% in addition to NLS signals for mitochondrial, extracellular, cytoplasmic, plasma membrane, or chloroplast localization were identified. Only five of 235 CaMYB proteins did not present NSL. Appendix A shows the protein list with at least one MYB repeat, and includes protein id, protein length, number of exons, NCBI annotation, id transcriptomic data, Pfam family domains, MYB classification, locus id, chromosome localization, strand orientation, isoelectric point, protein mass, and subcellular localization.

### 2.2. Phylogenetic Analysis of the MYB Family

We constructed a phylogenetic tree with 301 MYB proteins using the Neighbor joining method, based on the alignment of the MYB domain using the MEGA7 software, with a bootstrap test of 1000 replicates (Figure 1). We analyzed 126 CaMYBs, including all R2R3-MYB, 3R-MYB, atypical proteins (5R-MYB and CDC5-like), and three MYB-related proteins compared with the MYB family of *Arabidopsis thaliana* and 25 functionally characterized proteins from other plant species (Appendix A). The phylogenetic tree was consistent with the subgroups designated for *A. thaliana* MYB proteins [33,34]. Based on the phylogenetic tree topology, we grouped the CaMYB proteins in 46 clades (C1–C46), of which 37 clades were present both in *C. annuum* and *A. thaliana*. None of the analyzed CaMYBs was grouped with R2R3-MYB genes of the clades C31, C34, C36, and C42 of Arabidopsis. Conversely, clades C24, C25, C32, and C40 were grouped with only CaMYB proteins. The presence or absence of specie-specific MYBs could have resulted in functional divergence. In this vein, MYBs clustered in the same clade may have a conserved biological function. Based on the phylogenetic relationship of CaMYB proteins with known proteins of other species, we assigned putative functions to each subgroup (Appendix A). For instance, clade C1 comprised only 3R-MYB proteins of *C*. *annuum*, *A*. *thaliana,* and *Oryza sativa,* which were associated with the cell cycle process and stress response [35,36]. Clade C38 included several R2R3-MYBs: four from *C. annuum*, three from *A. thaliana,* and one from *Solanum lycopersicum*, all of which were related with the flavonol biosynthesis [37,38]. Clade C46 included three MYB-related genes from *C. annuum* and one of *Antirrhinum majus*, which was related to the flower development [39]. The 5R-MYB was found also in other *Capsicum* species (i.e., *C*. *chinense* and *C*. *baccatum*), but not in other Solanaceae genomes (i.e., *Solanum tuberosum*, *Solanum lycopersicum,* and *Nicotiana tabacum*). However, the CaMYB5R protein was clustered with 4R-MYB proteins whose functions are unknown.

### 2.3. Chromosomal Localization of the MYB Family

Based on the starting and ending position of *MYB* genes within the chromosomes, 213 genes with at least one MYB-domain were distributed among the 12 chromosomes of *C. annuum*, whereas 23 *MYB* genes stayed on as yet unmapped scaffolds (Figure 2 and Appendix A). Most of the *CaMYB* genes were concentrated at both extremes of the respective chromosome, with fewer exceptions of genes located in the middle section of the chromosome. Chromosome 3 encompassed the highest density of *CaMYB* genes (27), followed by chromosome 6 (26). Conversely, chromosome 8 (10) contained the lowest density of *MYB* genes. The distribution of *CaMYB* genes was not congruent with the MYB classification subgroups, except for a few gene pairs that had a close proximity localization on the chromosome. For example, *CaMYB36* and *CaMYB37* from Clade C16 were located only 0.009 Mb apart on chromosome 12. *CaMYB90* and *CaMYB91*, members of the clade C35, were located with a separation of 0.02 Mb on chromosome 11. Moreover, *CaMYB81* and *CaMYB82*, both belonging to clade C32, were localized on chromosome 1 separated by only 0.08 Mb.

### 2.4. Gene Structure Analysis

To explore the structural diversity of *CaMYB* genes, their exon-intron organization was analyzed. Most *CaMYBs* clustered in the same clade exhibited similar exon-intron structures (Figure 3), particularly regarding the number of exons, and in some cases the length of the exons and introns was also consistent. For instance, the members of clade C8 displayed a similar gene structure without introns and a similar exon size, or clade C21 in which all the elements showed three exons of similar size. Conversely, *3R-MYB* gene members of clade C1 presented multiple exons ranging 7–11 exons. The number of exons ranged from one to thirteen. The 39.83% of the *CaMYB* genes were organized by three exons, while 53.76% were organized with two or more than three exons, and 6.35% were intron-less.

### 2.5. Motif-Detection Analysis

The MEME motif-detection software [40] was used to analyze the diversification of the CaMYB proteins. We observed that the motif structure of MYB proteins was consistent with the classification of the clades based on the phylogenetic tree topology (Figure 4). Closer proteins revealed a better match in the motif arrangement. However, members of the same clade showed slight differences in the presence or absence of motifs outside the MYB domain, which may be related to whether or not these genes share a biological function in specific conditions. For example, in clade C25, CaMYB31 did not display motifs 7 and 9 unlike other MYBs in solanaceae, all of which exhibited one or both of these motifs, suggesting that the lack of motifs 7 and 9 could be related to the participation of *CaMYB31* gene in the capsaicinoid biosynthetic regulation given that this is a pathway unique to the *Capsicum* genus.

Additionally, MYB-related and MYB proteins with other domains (histones, ZZ, ARR, and SWI/SNF) were classified in subgroups based on their exon-intron structure and motif protein organization. MYB-related proteins were divided into 13 subgroups: DIVARICATA-like, RADIALIS-like, ETC-like, RVE/LHY/CCA-like, KAN-like, EFM/HH05-like, MYBC1/PCL-like, MYB1R/KUA-like, GLK-like, Atg14600-like, TRF-like, MYB-CC, and unknown genes (Appendix A). Each MYB-related subgroup shared a similar gene structure (Appendix A) and protein organization (Appendix A). For example, all *DIVARICATA-like* genes presented a gene structure composed of two exons, and a common MYB-binding domain with the SHAQKY consensus. Moreover, MYBC1/PCL-like shared an intron-less gene structure, and the MYB-binding domain included the SHLQKYR consensus. Furthermore, we classified MYB proteins with other domains in MYB-histone, MYB-ARR, MYB-ZZ, and MYB-SWI/SNF, which were highly conserved between *C. annuum* and *A. thaliana* species. Moreover, considering the conserved relationship between MYB proteins among plant species, we suggested a putative function for each subgroup (Appendix A).

### 2.6. Co-Expression Analysis

We generated RNA-seq data from flowers (0 dpa; days post-anthesis) and chili pepper fruits at 10, 20, 30, 40, 50, and 60 dpa to explore gene expression profiles (Figure 5). During the transcriptome assembly every gene received an identification number (ID) that represents the expression profile (Appendix A). We found that 218 of 235 *CaMYB* genes were expressed in at least one of the analyzed tissues. Fifteen of those *CaMYB* genes presented more than one ID, and some of them showed a distinct expression pattern (Figure 6). Additionally, 10.5% of *CaMYB* genes (*CaMYB7*, *CaMYB23*, *CaMYB28*, *CaMYB31*, *CaMY39*, *CaMYB44*, *CaMYB49*, *CaMYB50*, *CaMYB61*, *CaMYB62*, *CaMYB71*, *CaMYB78*, *CaMYB88*, *CaMYBR4*, *CaMYBR7*, *CaMYBR8*, *CaMYBR12*, *CaDIV8*, *CaDIV11*, *CaDIV13*, *CaDIV14*, *CaARR8*, and *CaMYB3R*-3) did not show expression in flower, while 4.1% (*CaMYB30*, *CaMYB37*, *CaMYB59*, *CaMYB66*, *CaMYB91*, *CaMYB99*, *CaMYBR3*, *CaMYBR5*, and *CaRL1*) were exclusively expressed in flower tissue. Furthermore, the co-expression analysis revealed thirteen main clusters of genes depending on their expression pattern through the development time points (Figure 6). For several genes, we found a correlation between its expression profile and its putative function. For example, *CaMYB17* exhibited a high expression in flower tissue, and showed phylogenetic proximity to *AtMYB125* that regulates pollen sperm cell differentiation [41] (Appendix A). Moreover, *CaMYB19*, *CaMYB20*, *CaMYB21,* and *CaMYB22*, all of them members of clade C11, were related to anther and tapetum development genes (subgroup S18 of Arabidopsis) [42,43], and displayed a high expression in flower tissue. However, we also found cases where the expression pattern did not correlate with any putative function, for example, *CaMYB12* and *CaMYB13* were related to subgroup S23 of Arabidopsis which were functionally related to pollen development [44], and they showed their highest expression at 50 dpa, with a lower expression in flower tissue. 

Furthermore, we looked for *CaMYB* genes that showed a positive correlation with key genes of important metabolic pathways, such as the phenylpropanoid, lignin, capsaicinoid, carotenoid, anthocyanin, and vitamin C biosynthetic pathways. Based on the profile expression, capsaicinoid and carotenoid biosynthetic genes were clustered by the metabolic pathway. *AT3* (*acyltranferase*), *Kas* (*ketoacyl-ACP synthase*), *pAmt* (*aminotransferase*), and *BCKDH* (*branched-chain amino acid transferase*), specific genes from the capsaicinoid biosynthetic pathway, presented the characteristic known expression profile that was null in flowers, low at 10 dpa, increased to a maximum at 20 dpa, decreased at 30–40 dpa, and null between 50–60 dpa. These genes were clustered with *CaMYB31*, previously functionally characterized as the capsaicinoid biosynthesis regulator [27,45], and also co-expressed with *CaRVE4*, *CaARR4*, *CaMYB115*, and *CaMYB103*. Surprisingly, *CaMYB48* and *CaMYB108* recently identified as regulators of capsaicinoid biosynthesis [28,29] did not exhibit a positive correlation with the capsaicinoid structural genes. *CCS* (*capsanthin-capsorubin synthase*), *BCH* (*β-carotene hydroxylase*), and *PSY* (*phytoene synthase*), carotenoid biosynthetic genes, showed a lower or even null expression between 0–40 dpa, with a sudden increase between 50–60 dpa. Interestingly, these genes were clustered with six *MYB*-related genes (*CaDIV1*, *CaDIV3*, *CaMYBR13*, *CaTRF2*, *CaMYBC1*, and *CaPHR9*) and an atypical *MYB* (*CaMYB5R*). Contrariwise, structural genes of the anthocyanin [*DFR* (*dihydroflavonol 4-reductase*), *F3′5′H* (flavonoid 3′, 5′-hydroxylase), and *CHS* (*chalcone synthase*)], phenylpropanoid [*PAL* (*phenylalanine ammonia lyase*), *4CL* (*4-coumarate-CoA ligase*) and *C4H* (*cinnamic acid 4-hydroxylase*)], vitamin C [*GLDH* (*L-galactono-1,4-lactone dehydrogenase*), *GalDH* (*L-galactose-1-dehydrogenase*) and *GME* (*GDP-D-mannose-3′,5′-epimerase*)], and lignin [*CAD* (*cinnamyl alcohol dehydrogenase*), *CCR* (*cinnamoyl CoA reductase*) and *POD* (*peroxidase*)] biosynthetic pathways were not clustered by the metabolic pathway. Nevertheless, the expression profile of these genes was highly correlated with at least one *CaMYB* gene. *F3′5′H* was grouped with *CaMYB54*, *CaMYB67*, *CaMYB69*, and *CaMYB90*, all of them *R2R3-MYB* genes highly expressed at 10 dpa. *DFR* was highly expressed at 40 dpa and clustered with two R2R3-MYB genes (*CaMYB25* and *CaMYB46)*, and one *DIVARICATA*-like gene (*CaDIV11*). *PAL*, *CHS,* and *POD* presented the highest expression in flowers, and suddenly decreased during the fruit development. These genes co-expressed with a large cluster of *R2R3-MYB* genes (*CaBLIND, CaMYB11*, *CaMYB24***,**
*CaMYB30*, *CaMYB32*, *CaMYB33*, *CaMYB34*, *CaMYB37*, *CaMYB57*, *CaMYB59*, *CaMYB66*, *CaMYB75*, *CaMYB85*, *CaMYB89*, *CaMYB91*, *CaMYB99*, *CaMYB102*, and *CaMYB105*) and *MYB*-related genes (*CaKAN7*, *CaKUA4*, *CaRL1*, *CaRL2*, *CaMYBR3*, *CaMYBR5*, and *CaPHR3*). *4CL* displayed its highest expression in flower, and gradually decreased during the fruit development along with *CaARR7* and *CaKUA3*. *C4H* exhibited its highest expression in flower that was slowly decreasing during the fruit development, and it was co-expressed with *CaDIV4*. Regarding the expression of genes involved in vitamin C biosynthesis, *GLDH* displayed a steady expression throughout the fruit development, and it correlated closely to *CaA*. *GalDH* showed the highest expression between 10–30 dpa, along with *CaMYB1*, *CaMYB43*, and *CaMYB91*. *GME* (vitamin C biosynthesis) and *CAD* (lignin biosynthesis) showed the highest expression in flower tissue, then their expression suddenly decreased between 10–40 dpa, and slightly increased at 50–60 dpa. Both genes were closer to *CaMYB36*, *CaMYB64*, *CaMYB80*, *CaMYB98*, *CaMYB108, CaLHY*, *CaDIV7*, *CaDIV10*, *CaSWI3B*, *CaKAN6*, and *CaKUA2* expression. *CCR* started accumulating at 0–10 dpa, peaked at 20 dpa, and abruptly decreased at 30–40 dpa, and again its expression increased at 50–60 dpa. The CCR expression showed a positive correlation with *CaMYB37* and an isoform of *CaLHY*.

### 2.7. RT-qPCR Analysis

To identify *CaMYB* genes as regulators of important secondary metabolic pathways, we analyzed the expression profile of *AT3* (capsaicinoid biosynthesis), *CCS* (carotenoid biosynthesis), *PAL* (phenylpropanoid biosynthesis), *CAD* (lignin biosynthesis), and *GLDH* (vitamin C biosynthesis) structural genes compared with their possible *CaMYB* regulators throughout the chili pepper fruit development (Figure 7). The transcript FPKM expression was similar to the quantitative relative expression. The *AT3* expression was not detected in flower tissue, very low at 10 dpa, increased to a maximum between 20–30 dpa, decreased at 40 dpa, and was undetectable at 50 and 60 dpa. We verified that the *CaMYB31* expression positively correlated with *AT3*. Moreover, the expression profile of *CaMYB103*, *CaMYB115*, and *CaDIV14* positively correlated with that of *AT3* and *CaMYB31*, suggesting that these transcription factors could be involved in the regulation of the capsaicinoid biosynthesis pathway. The *CCS* expression pattern was low between 0 and 40 dpa, and highly expressed at 50 and 60 dpa. The *CaDIV1* and *CaMYB3R*-5 expression profile was correlated with that of *CCS*, being good candidates to regulate the *CCS* transcription. *PAL* showed the highest expression level at 0 dpa, and gradually decreased during the fruit development. Based on the RNA-seq data, several *CaMYB* genes were positively correlated with *PAL*, of which *CaMYB32*, *CaMYB33*, and *CaMYB93* expression pattern was investigated. These three *CaMYB* genes showed the highest expression at 0 dpa, whereas the *CaMYB93* expression gradually decreased during the fruit development, *CaMYB32* and *CaMYB33* presented a very low or even null expression during the fruit development. The *CAD* expression was highest at 0 dpa, drastically decreased between 10–20 dpa, diminished a little more between 30–40 dpa, and increased moderately between 50–60 dpa. We verified the expression profile of two possible candidates to regulate the expression of *CAD*, *CaMYB98,* and *CaMYB108*, and both presented a positive correlation with *CAD*. The *GLDH* expression showed the highest value between 0–10 dpa, diminished at 20 dpa, and remained constant until 60 dpa. *CaMYB16* presented an expression pattern similar to *GLDH*.

## 3. Discussion

The MYB family, one of the largest transcription factor families, has been implicated in diverse important biological process in plants such as primary and secondary metabolism, developmental processes, biotic and abiotic stress response, cellular and organ morphogenesis, and cell cycle control [3,8]. MYB transcription factors have been identified in several plant species such as Arabidopsis, rice, potato, pineapple, and tomato [17,18,19,21,34]. In the current study, we performed a wide analysis of *CaMYB* genes family in chili pepper in comparison to the *MYB* gene family of *A. thaliana* and other known plant *MYB* genes.

### 3.1. The 235 CaMYB Genes Were Identified in Capsicum annuum Genome

A total of 235 non-redundant genes with at least one MYB repeat were identified in the *C. annuum* genome. These genes were divided into five subfamilies, including 116 R2R3-MYB, five 3R-MYB, 92 MYB-related, two atypical MYBs (CaMYB5R and CaMYBCDC), and 20 MYB proteins with other conserved domains (Appendix A). Recently, Wang et al. [31] identified 108 R2R3-MYBs in chili pepper, five genes less than in our study, probably since our strategy for the identification and classification of CaMYBs was also based on the comparative analysis with other known MYB plant proteins. Consistent with our results, the R2R3-MYB subfamily has been reported as the largest subfamily of the MYB family in other plant species [15,17,46], with some exceptions [21,47]. The total number of *CaMYB* genes was higher than the number of *MYB* genes identified in Arabidopsis (198), tomato (127), and pineapple (184), but lower than those in potato (253), sunflower (245), and sesame (287) [17,18,19,20,21,47], suggesting a distinct degree of evolutionary expansion of the MYB family among plant species. The molecular weight and isoelectric points play important roles in determining the molecular and biochemical function [48]. We investigated CaMYB protein sizes and their pI, which presented a clear variation, probably due to their roles across different environments, contributing to a great functional diversity in MYB proteins. A typical TF contains a DNA-binding region, an oligomerization site, a transcription-regulation domain, and a nuclear localization site [1]. To inspect whether all CaMYBs presented a NLS, we predicted the subcellular localization of CaMYB proteins, of which almost 98% exhibited nuclear localization. Those genes that did not have an NLS were two *MYB*-related genes (CaPHR4 and CaRL3) and three *MYBs* with other conserved domains (CaSWI3A, CaMYBH4, and CaMYBH5). Transcriptional regulatory proteins without NLS may be imported into the nucleus by dimerization with proteins that do contain NLS [49]. *CaMYBs* were distributed throughout all twelve chromosomes of chili pepper, but their distribution seemed to be uneven, showing the highest density on the top and bottom of the chromosomes (Figure 2). This *MYB* gene distribution was similar to previous studies in other Solanaceae members, such as tomato [19] and potato [21].

### 3.2. CaMYB Family Relationships with Other Plant MYB Proteins

A phylogenetic tree was constructed with the R2R3-MYB, 3R-MYB, and atypical proteins of *C. annuum* compared with the MYB family of Arabidopsis and other known function MYB proteins (Figure 1). The phylogenetic analysis was congruent with previous reports for Arabidopsis [33,34]. Genes grouped in the same clade may have a common ancestor and thus share conserved functions. Most of the CaMYBs were clustered with AtMYBs, providing interesting insights on the roles of *CaMYB* genes (Appendix A). For example, CaMYB97, CaMYB98, CaMYB99, and CaMYB100 grouped together with AtMYB11, AtMYB12, AtMYB111, and SlMYB12 could be implicated in the regulation of flavonoid biosynthesis [37,38,50]. Clade C15 was constituted by CaMYB32 and CaMYB33 of *C*. *annuum* and the well-known proteins AtMYB21, AtMYB24, AtMYB57, AmMYB305, and AmMYB340 that have been implicated in the control of *PAL* gene and stamen filament elongation [51,52]. Conversely, some CaMYBs did not cluster with any AtMYB proteins, suggesting evolutive events of the gain or loss of genes. The biochemical and physiological differences between *C. annuum* and *A. thaliana* may hint at the existence of species-specific MYBs with specialized functions. For instance, *CaMYB31* did not exhibit orthologous genes in Arabidopsis, probably since *CaMYB31* regulates the capsaicinoid biosynthetic pathway, which is specific to the *Capsicum* genus. Likewise, clade C34 did not include CaMYBs, but only AtMYBs, which are involved in the glucosinolate biosynthetic process, which is predominant in the Brassicaceae family.

A subgroup classification was highly supported by the gene structure (Figure 3) and motif protein arrangement analysis (Figure 4). Consistent with previous studies [53,54], most of the *MYB* genes within the same subgroup shared a similar exon-intron structure, showing that *MYB* genes are highly conserved among species. MYB proteins comprise the conserved MYB domain that recognizes the promoter of the target gene, and a highly variable region responsible for the regulatory activity [3,8]. The motif analysis revealed that CaMYB proteins belonging to the same subgroup displayed common motifs in the amino acid sequence outside the MYB domains, suggesting that they might share similar functions. Clade C33, which included four *CaMYB* genes along with subgroup S11 of Arabidopsis, shared three exons of very similar size and they have been implicated in salt-tolerance and suberin biosynthesis [55,56]. In addition, these proteins showed conserved motifs in the C-terminal region (i.e., motifs 5 and 6) indicating common functions. However, they also displayed exclusive motifs for CaMYBs (i.e., motif 8) and AtMYBs (i.e., motif 10).

Furthermore, we classified and assigned putative functions to MYB-related and MYBs with other conserved domains based on their intron-exon organizations and motif protein composition, which was highly conserved (Appendix A). The subgroup KAN-like contained seven *CaMYB*-related genes that shared a MYB-domain encompassing the SHLQMYR consensus. The exon-intron organization and motif protein arrangement revealed that *CaKAN1*, *CaKAN2*, *CaKAN3,* and *CaKAN4* exhibited similarities with *AtKAN2* suggesting that they may be involved in lateral organ formation [57]. Conversely, *CaKAN5*, *CaKAN6,* and *CaKAN7* were more distant to *AtKAN2*, thus suggesting they may be playing other roles in *Capsicum*. The EFM/HHO5-like subgroup included four *CaMYB*-related genes together with *AtEFM* and *AtHHO5* genes, in which all presented a MYB-domain with the SHLQKYR consensus. Outside the MYB-domain, those proteins exhibited motifs 2 and 6, implying common possible functions in the flowering process [12,14].

### 3.3. CaMYB Gene Expression Profiles during the Chili Pepper Fruit Development

The transcriptome annotation of the CaMYB family provides insights into their functions. A comparative expression analysis has allowed identifying numerous regulatory genes important for the control of different biological processes in *Capsicum* [25,27,30]. We performed the co-expression analysis of *CaMYB* genes using transcriptome data of flower (0 dpa) and chili pepper fruit throughout different developmental stages (10–60 dpa). We detected transcripts for 218 *CaMYB* genes (92.7%) in at least one of the tissues tested, and they showed different tissue-specific expression patterns, suggesting that these genes might be implicated in the control of diverse biological processes related to the specific expression pattern that they showed in the corresponding tissues. Most of the *CaMYB* genes in the same subgroup exhibited dissimilar expression profiles suggesting that these genes may have similar roles in the distinct tissues or under different environmental conditions. For the heat map construction, in addition to including the *CaMYB* genes, we also analyzed key structural genes of the phenylpropanoid, lignin, capsaicinoid, carotenoid, anthocyanin, and vitamin C biosynthetic pathways to identify *CaMYB* genes as regulator candidates of such pathways. The co-expression analysis showed that at least one *CaMYB* gene positively correlated with the biosynthetic genes analyzed (Figure 6). Moreover, the expression profile for some of these candidates was verified by RT-qPCR (Figure 7), reinforcing that they could be strong candidates to regulate these metabolic routes.

The phenylpropanoid metabolism generates a great range of secondary metabolites that contribute to numerous biological processes, such as plant growth and development, and biotic and abiotic response [58]. PAL is responsible for the first step in the synthesis of phenylpropanoid-derived compounds. Among all the *CaMYB* genes that positively correlated with the *PAL* expression profile, we corroborated that *CaMYB32*, *CaMYB33*, and *CaMYB93* expression patterns effectively co-expressed with the *PAL* gene. Unlike the RNA-seq analysis, the RT-qPCR study showed that the *PAL* gene presented the highest expression in flowers (0 dpa), and then gradually decreased throughout the fruit development, probably due to the wide diversification of the phenylpropanoid pathway (i.e., flavonoid, lignin, capsaicinoid, etc.), which is also carried out in the different developmental stages of chili pepper fruit. The *CaMYB93* expression correlated quite well with the *PAL* expression profile throughout the fruit development, and based on the phylogeny tree, CaMYB93 grouped with subgroup S1 implicated in secondary metabolism, plant development, and stress response [59,60,61,62,63]. Therefore, the *CaMYB93* gene may be regulating *PAL* transcription throughout the fruit development. Contrarily, *CaMYB32* and *CaMYB33* showed the highest expression in flower tissue, very low in chili pepper fruit at 10–20 dpa, and undetectable at 30–60 dpa. These two TFs were the R2R3-MYB type and phylogenetically related to AmMYB305, which has been proposed to activate the expression of phenylpropanoid biosynthetic genes in flowers [51], and with AtMYB21, which has been reported to be required for the activation of *PAL* [64]. Additionally, it is known that members of the subgroup S19 are expressed primarily in flowers [65]. Based on these results, both R2R3-MYB TFs are strong candidates to regulate the *PAL* gene expression in both flower tissue and early chili fruit developmental stages. Moreover, these genes shared a highly conserved motif structure in the C-terminal region with the S19 of Arabidopsis and AmMYB305 and AmMYB340 proteins, supporting the fact that they play a common role in the phenylpropanoid regulation.

The most widely known characteristic of chili pepper fruits is their capacity to produce capsaicinoids which are responsible for the pungency sensation [66]. To date, three MYBs transcription factors (*CaMYB31*, *CaMYB48,* and *CaMYB108*) have been reported as regulators of the capsaicinoid biosynthesis pathway [27,28,29]. In a recent work, six candidate *CaR2R3-MYB* genes were proposed to be regulating the synthesis of capsaicinoids: Capana08g001690, Capana02g003351, Capana08g000900, Capana02g000906, Capana01g000495, and Capana07g001604 [31] which correspond to CaMYB47, CaMYB87, CaMYB73, CaMYB92, CaMYB74, and CaMYB64, respectively (Appendix A). In our co-expression analysis, only the *CaMYB31* expression profile correlated positively with the expression profiles of capsaicinoid biosynthetic marker genes. To identify more *CaMYB* gene candidates possibly involved in the regulation of the capsaicinoid biosynthetic pathway, *AT3* and *CaMYB31* expression was quantified and compared with two *R2R3*-*MYB* (*CaMYB103* and *CaMYB115*) and one *MYB*-related (*CaDIV14*) genes, resulting in a positive correlation, suggesting that they are strong candidates to regulate the capsaicinoid accumulation. These *CaMYB* genes were phylogenetically grouped in different clades, thus they could be involved in different functional activities necessary for the capsaicin production, such as molecular, biochemical, or physiological processes. *CaMYB103* did not show any orthologous genes in Arabidopsis, indicating perhaps a specialized role in *Capsicum*, such as the control of capsaicinoid biosynthesis. *CaMYB115* was grouped with genes related to the phenylpropanoid pathway [67], thus this gene may be implicated in the regulation of early genes of capsaicinoid biosynthesis. Conversely, *CaDIV14* was clustered with *DIVARICATA*-like genes that have been characterized for their participation in the flower development [39]. Since *CaDIV14* did not show an expression in flower tissue, but it did exhibit the characteristic expression pattern for capsaicinoid biosynthetic genes, it is likely that this gene could be associated with the blister (capsaicin deposit) development process, which has been linked to the capsaicinoid accumulation [68,69].

Lignin is one of the main components of plants cell wall and contributes to plant growth, tissue/organ development, and response to various stresses [70]. The regulation of the lignin biosynthesis pathway has not been well described in *Capsicum*. The content of lignin in different developmental stages of chili pepper fruit was determined, resulting in a gradual decrease in lignin content from 14 until 42 dpa [71,72]. Studies have demonstrated that the lignin biosynthesis is regulated by MYB transcription factors [73,74]. To explore possible MYB regulators of the lignin biosynthetic pathway, we analyzed the relative expression of the *CAD* gene, responsible for the last step in the biosynthesis of lignins. We confirmed that *CaMYB98* and *CaMYB108* genes co-expressed with the *CAD* gene expression during the development of chili pepper fruit. Interestingly, the expression profile was opposite to that of capsaicinoid biosynthetic genes, probably due to the precursor competition between the capsaicinoid and lignin production [71,72]. Unexpectedly, the *CaMYB108* expression profile was not consistent with previous studies that have suggested that this gene regulates the capsaicinoid biosynthesis pathway [28]. Both TFs were phylogenetically related to MYB proteins involved in flavonol biosynthesis, overall growth, and response to biotic and abiotic stresses, which is consistent with some biological roles of lignins.

Chili pepper fruits also synthetize and accumulate carotenoid pigments, which are responsible for the yellow, orange, and red colors. The *CCS* gene is responsible for the last biochemical step of capsanthin and capsorubin biosynthesis, which confer the red color [75]. The *CCS* gene was used here as a reference to identify *CaMYB* genes candidates involved in the regulation of the carotenoid biosynthesis pathway. *CaDIV1* and *CaMYB3R*-5 genes shared a similar expression profile during the fruit development as well as with the *CCS* gene. *CaDIV1* belongs to the *DIVARICATA*-like genes classification (R-R type MYB domain) that to date have been characterized by their participation in plant development and stress response [13]. *CaMYB3R*-5 was classified as 3R-MYB, whose function has been related to the cell cycle process and abiotic stress response [76]. The temporal and spatial regulation of cell proliferation impacts on the shape and size of a plant organ in response to specific environmental conditions or developmental stages [77]. *CaDIV1* and *CaMYB3R*-5 genes might be involved in the regulation of organelle structures biogenesis for carotenoid accumulation.

The L-ascorbic acid (vitamin C) is the most abundant antioxidant in plant cells, which participates in diverse biological processes, including cell expansion, environment response, photoprotection, and photosynthesis, etc. [78]. GLDH is responsible for the last step in producing vitamin C through the L-galactose pathway, and it was used here as a reference gene to find *CaMYB* genes candidates possibly involved in the control of vitamin C biosynthesis. The *CaMYB16* expression profile correlated positively with both the *GLDH* expression pattern and content of vitamin C throughout the chili pepper fruit development [78]. The constant production of vitamin C during the chili pepper fruit development can be necessary either for protection purposes or development. Based on the phylogenetic analysis, *CaMYB16* was closer to the subgroup S22 of Arabidopsis, whose elements are implicated in biotic and abiotic responses, and plant growth and development, which are considered biological functions of vitamin C in plants. Taking it all together, this study might contribute as an important resource to propose new *MYB* gene candidates for the regulation of diverse biological processes in *Capsicum* spp.

## 4. Materials and Methods

### 4.1. Plant Material and Growth Conditions

Chili pepper (*Capsicum annuum*) “Tampiqueño 74” (Serrano type) were grown under greenhouse conditions and fertilized with a Long Ashton solution every 2 weeks. Flowers (0 dpa) and whole chili pepper fruits at 10, 20, 30, 40, 50, and 60 dpa were harvested from at least 12 plants, immediately frozen in liquid nitrogen, and stored at −80 °C. These materials were used for the RNA-seq library construction and RT-qPCR expression analysis.

### 4.2. Identification of MYB Proteins

The genome database of chili pepper cultivar Zunla-1 (*C*. *annuum*) was used in this study [24]. The identification of MYB genes was carried out with a Blastp search using as query the Pfam domains: PF00249 and PF1392. To verify the presence of the significant MYB domain, all the sequences were examined using the Pfam database (https://pfam.xfam.org, accessed on 5 January 2021) [32]. The protein, coding and genome sequences, and chromosome localization were downloaded from the RefSeq_protein National Center for Biotechnology Information database. Three hundred and forty-five proteins with at least one MYB domain were identified, which were localized in 235 loci. Therefore, 235 non-redundant CaMYBs were identified in *C. annuum*.

### 4.3. Phylogenetic Analysis and Sequence Analysis

For the phylogenetic tree construction, we selected 126 non-redundant CaMYB proteins, including all R2R3-MYB, 3R-MYB, atypical proteins (CaMYB5R and CaMYBCDC), and three MYB-related proteins. CaMYBs were compared with the MYB family of *Arabidopsis thaliana* and 25 MYB proteins were functionally characterized from other plant species (Appendix A). The sequences were aligned with ClustalW using default parameters, and it was manually adjusted. Based on the alignment of the MYB domain, a phylogenetic tree was constructed with the Neighbor-joining method, model JTT+G, and a bootstrap test of 1000 replicates using the MEGA7 software. The isoelectric point and protein molecular weight were predicted using the isoelectric point calculator (IPC) software (http://isoelectric.org/index.html, accessed on 5 January 2021) [79]. To predict the subcelullar localization of CaMYBs, the CELLO v.2.5: Subcellular localization predictor (http://cello.life.nctu.edu.tw, accessed on 5 January 2021) [80] was used. MEME version 5.2.0 (http://meme-suite.org/tools/meme, accessed on 5 January 2021) [40] was used to discover conserved motifs outside the MYB domain. The exon-intron structures of *CaMYB* genes were schemed by the Gene Structure Display Server version 2.0 (GSDS) (http://gsds.gao-lab.org, accessed on 5 January 2021) [81] comparing the CDS and genome sequences. The chromosomal distribution of *CaMYB* genes was mapped using the Map Gene 2 Chromosome (MG2C) server version 2.0 (http://mg2c.iask.in/mg2c_v2.0/, accessed on 5 January 2021) [82]. The CaMYB subfamily classification was carried out according to the topology of the phylogenetic tree, motif protein composition, and exon-intron organization.

### 4.4. RNA-Seq Library Construction and Processing

Total RNA was extracted from flowers (0 dpa) and whole chili pepper fruits at 10, 20, 30, 40, 50, and 60 dpa using a NucleoSpin^TM^ RNA plant kit (MACHEREY-NAGEL, Bethlehem, PA, USA) following the manufacturer’s indications. The extraction was performed in duplicate for each sample. Flower samples were collected from 6–12 different plants, and fruit samples included 3–6 different plants. The RNA quality was verified by 1% agarose gel electrophoresis and the RNA integrity number (RIN) for each sample was determined. For library construction, sequencing and mapping to a reference genome, the Novogene company (Sacramento, CA, USA) services were used. Libraries were prepared and sequenced using the Illumina Platform to obtain 6G raw paired-end reads of 250–300 pb per sample. These reads were mapped to the *Capsicum* reference genome, identified by the protein product, and annotated with gene ontology (GO) and Kyoto encyclopedia of genes and genomes (KEGG) terms.

### 4.5. Co-Expression Analysis

The heat map was generated in R Studio version 1.2.5019 using the pheatmap package version 1.0.12 with the FPKM expression data. The genes were clustered based on the Pearson correlation analysis.

### 4.6. Quantitative PCR Assays

Total RNA was extracted, purified, and treated with DNase I (MACHEREY-NAGEL, Bethlehem, PA, USA) according to the manufacturer’s instructions. The RNA extraction from flowers and whole chili pepper fruits at different developmental stages was performed in triplicate experiments for each sample. The cDNA was synthesized with SuperScript III reverse transcriptase (Invitrogen, Carlsbad, CA, USA) and adjusted to 100 ng μL^−1^. The PCR primers were designed to avoid the conserved region and to amplify products of 90 to 142 bp (Appendix A). We analyzed the expression of eleven *CaMYB* genes and five key structural genes of the phenylpropanoid, lignin, capsaicinoid, carotenoid, and vitamin C biosynthesis pathways. The *EF1-α* elongation was used as the normalization reference gene. RT-qPCR assays were performed as reported by Arce-Rodríguez and Ochoa-Alejo [83].

## 5. Conclusions

The characterization and classification of gene families is a crucial first step for functional studies. A total of 235 *CaMYB* genes were identified and classified, comprising R2R3-MYB, 3R-MYB, atypical MYBs, and MYB-related genes. These genes were unevenly distributed on the twelve chromosomes of *C. annuum*. Based on the phylogenetic relationships, most of the CaMYBs presented possible orthologous in Arabidopsis, indicating common evolutionary origins. Furthermore, a computational analysis revealed that *CaMYB* genes might play roles in diverse biological processes, both conserved and specialized functions. The co-expression analysis in flower and fruits at different developmental stages showed that *CaMYB* genes were differentially expressed in all the tissues analyzed, supporting the idea that CaMYBs are functionally divergent. The integration of our results allowed us to propose some strong MYB candidates that might be involved in the regulation of phenylpropanoid, lignin, capsaicinoid, carotenoid, and vitamin C biosynthesis, providing new insights into the role of MYB transcription factors in secondary metabolism. Further functional characterization of *CaMYB* genes is needed for a better understanding of the role and regulatory mechanisms of the MYB family in *Capsicum* spp.

## Figures and Tables

**Figure 1 ijms-22-02229-f001:**
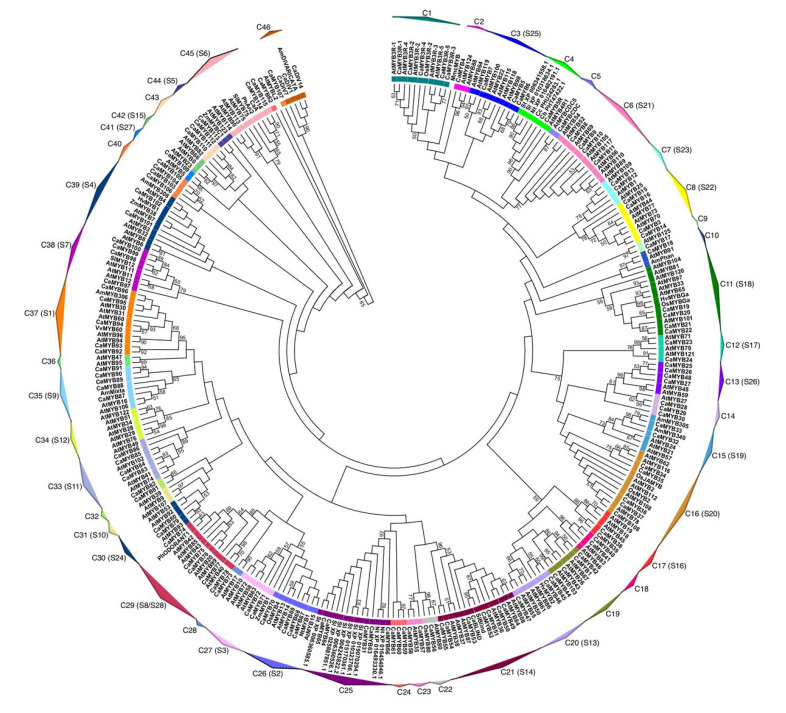
Phylogenetic relationship of CaMYB proteins with other plant MYB transcription factors. A phylogenetic tree was constructed using the Neighbor-joining method based on the MYB domain alignment using the MEGA7 software, with 1000 bootstrap replicates. The tree shows the 46 clades (C1–C46) with a high bootstrap value (highlighted with colored squares for each clade). Bootstrap values >50 are indicated on the nodes. Ca: *Capsicum annuum*; Cb: *Capsicum baccatum*; Cc: *Capsicum chinense*; At: *Arabidopsis thaliana*; Sl: *Solanum lycopersicum*; St: *Solanum tuberosum*; Nt: *Nicotiana tabacum*; Am: *Antirrhinum majus*; Os: *Oryza sativa*; Mm: *Mus musculus*; Hv: *Hordeum vulgare*; Zm: *Zea mays*; Ph: *Petunia hybrida*; Vv: *Vitis vinifera*. Accession numbers for all protein sequences are listed in Appendix A.

**Figure 2 ijms-22-02229-f002:**
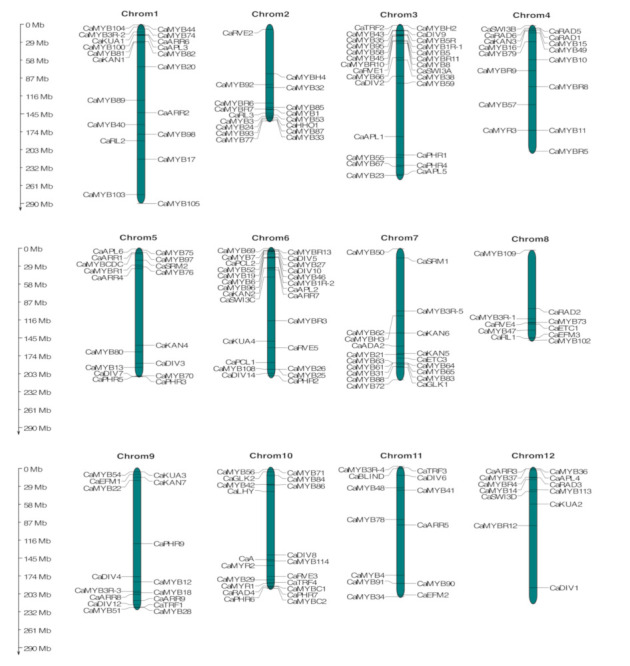
Chromosomal distribution of *CaMYB* genes. The chromosomal position was mapped according to the *Capsicum annuum* genome. Only 213 MYBs were mapped to the 12 chromosomes of *C. annuum*. The chromosome number is indicated at the top of each chromosome. The scale on the left is in megabases (Mb).

**Figure 3 ijms-22-02229-f003:**
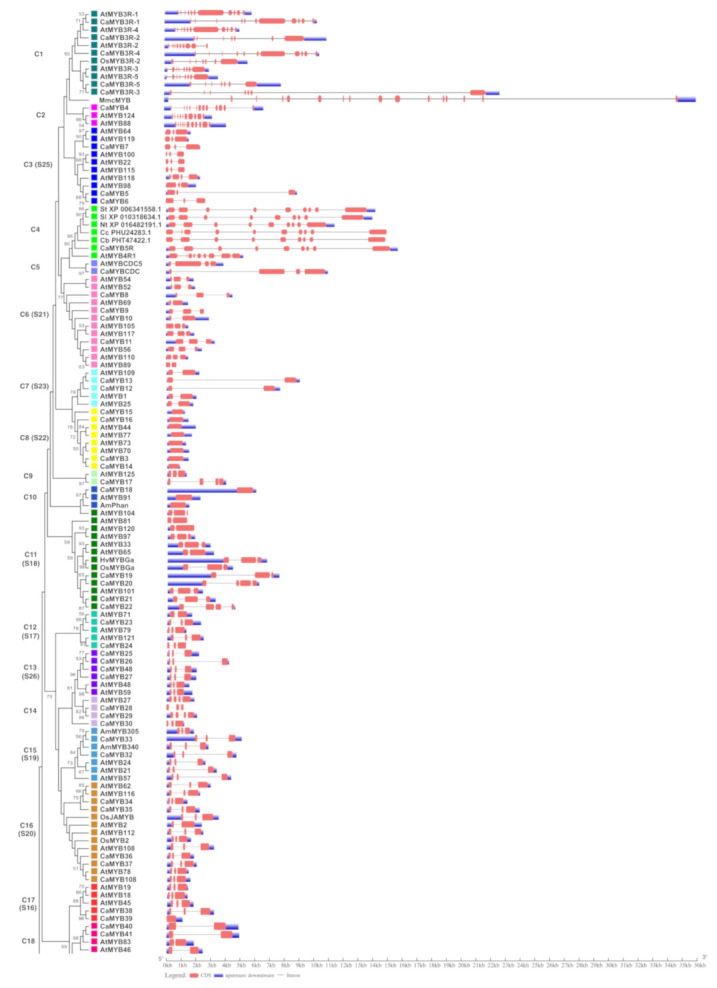
Exon-intron structures of *CaMYB* genes. Red bars represent exons and solid lines correspond to introns. Upstream and downstream regions are indicated by the blue boxes. The scale on the bottom is in kilobases (Kb). Ca: *Capsicum annuum*; Cb: *Capsicum baccatum*; Cc: *Capsicum chinense*; At: *Arabidopsis thaliana*; Sl: *Solanum lycopersicum*; St: *Solanum tuberosum*; Nt: *Nicotiana tabacum*; Am: *Antirrhinum majus*; Os: *Oryza sativa*; Mm: *Mus musculus*; Hv: *Hordeum vulgare*; Zm: *Zea mays*; Ph: *Petunia hybrida*; Vv: *Vitis vinifera*. The protein IDs are listed in Appendix A.

**Figure 4 ijms-22-02229-f004:**
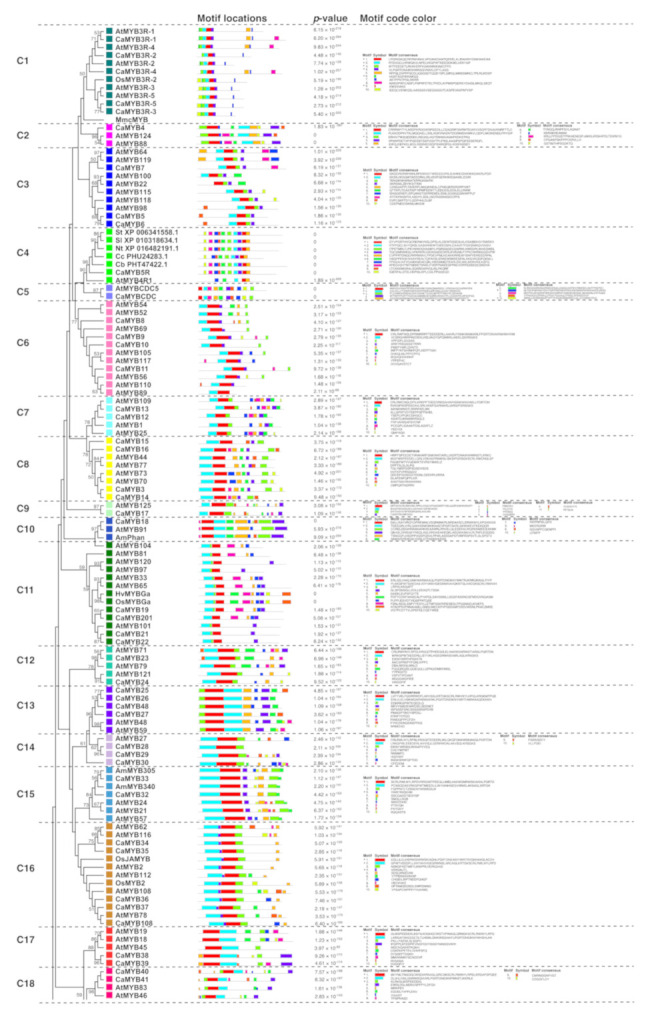
Schematic representation of motif composition of CaMYB subgroups. Each subgroup was analyzed using the MEME motif-detection software. The length of the solid line represents the length of the protein sequences. Colored boxes represent different motifs. Asterisks (*) indicate motifs containing the conserved MYB domain sequence. Ca: *Capsicum annuum*; Cb: *Capsicum baccatum*; Cc: *Capsicum chinense*; At: *Arabidopsis thaliana*; Sl: *Solanum lycopersicum*; St: *Solanum tuberosum*; Nt: *Nicotiana tabacum*; Am: *Antirrhinum majus*; Os: *Oryza sativa*; Mm: *Mus musculus*; Hv: *Hordeum vulgare*; Zm: *Zea mays*; Ph: *Petunia hybrida*; Vv: *Vitis vinifera*. The protein IDs are listed in Appendix A.

**Figure 5 ijms-22-02229-f005:**
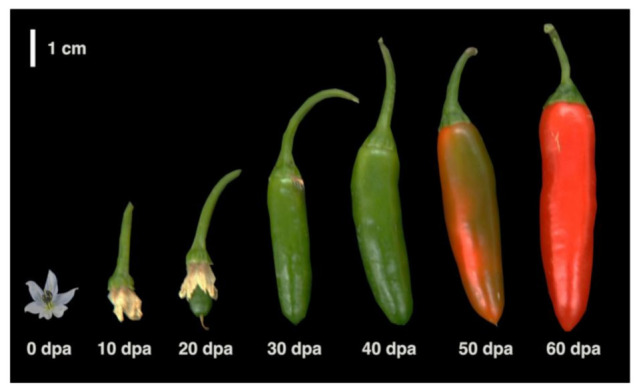
Serrano “Tampiqueño 74” fruits throughout developmental stages. Flower (0 dpa; days post-anthesis) and chili pepper fruits at 10, 20, 30, 40, 50, and 60 dpa.

**Figure 6 ijms-22-02229-f006:**
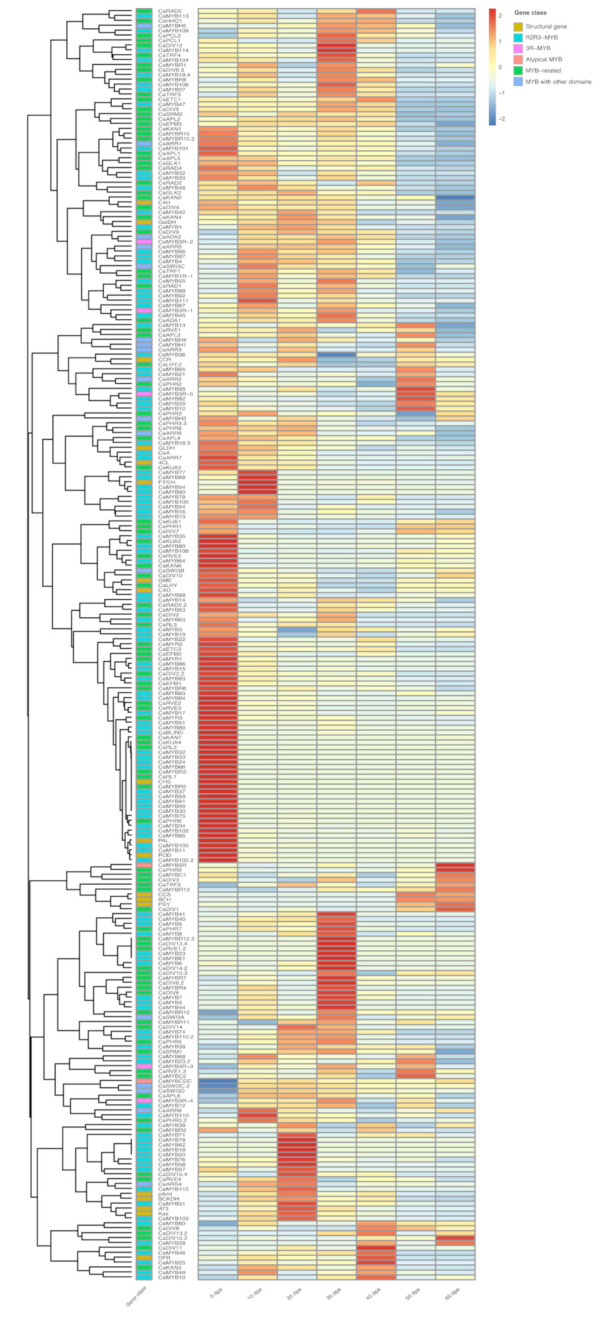
Expression profiles of *CaMYB* genes in flowers and during chili pepper fruit development. The heat map shows the expression pattern of *CaMYB* genes in flower (0 dpa; days post-anthesis) and in chili pepper fruit at 10, 20, 30, 40, 50, and 60 dpa in FPKM (fragments per kilobase of transcript sequence per millions base pairs sequenced). Red color represents the highest relative expression level. The number extension (0.2, 0.3, and 0.4) indicates *CaMYB* genes with more than one expression profile. Capsaicinoid biosynthetic genes: *AT3* (*acyltranferase*), *Kas* (*ketoacyl-ACP synthase*), *pAmt* (*aminotransferase*), and *BCKDH* (*branched-chain amino acid transferase*); carotenoid biosynthetic genes: *CCS* (*capsanthin-capsorubin synthase*), *BCH* (*β-carotene hydroxylase*), and *PSY* (*phytoene synthase*); anthocyanin biosynthetic genes: *DFR* (*dihydroflavonol 4-reductase*), *F3′5′H* (flavonoid 3′, 5′-hydroxylase), and *CHS* (*chalcone synthase*); phenylpropanoid biosynthetic genes: *PAL* (*phenylalanine ammonia lyase*), *4CL* (*4-coumarate-CoA ligase*), and *C4H* (*cinnamic acid 4-hydroxylase*); vitamin C biosynthetic genes: *GLDH* (*L-galactono-1,4-lactone dehydrogenase*), *GalDH* (*L-galactose-1-dehydrogenase*), and *GME* (*GDP-D-mannose-3′,5′-epimerase*); lignin biosynthetic genes: *CAD* (*cinnamyl alcohol dehydrogenase*), *CCR* (*cinnamoyl CoA reductase*), and *POD* (*peroxidase*).

**Figure 7 ijms-22-02229-f007:**
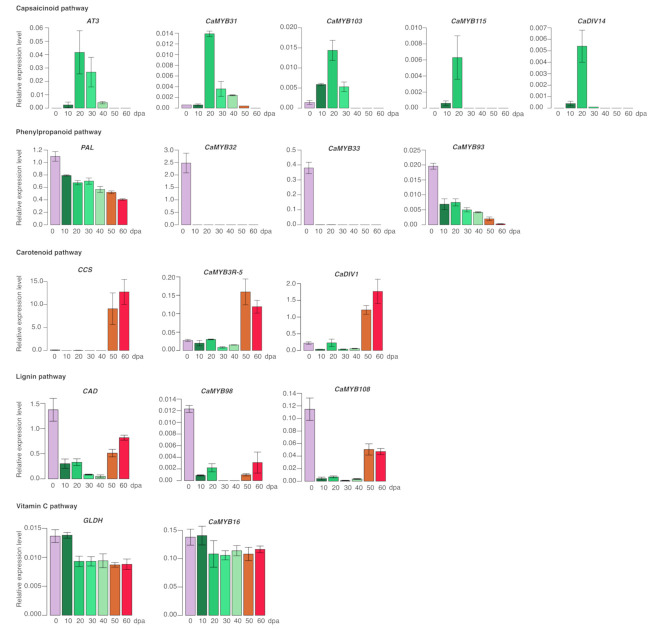
RT-qPCR of the expression profile of *AT3* (capsaicinoid pathway), *PAL* (phenylpropanoid pathway), *CCS* (carotenoid pathway), *CAD* (lignin pathway), and *GLDH* (vitamin C pathway) structural genes and their putative *CaMYB* regulators. The relative expression level was analyzed in flower (0 dpa; days post-anthesis) and in chili pepper fruit at 10, 20, 30, 40, 50, and 60 dpa. *AT3*: *Acyltranferase*; *PAL*: *Phenylalanine ammonia lyase*; *CCS*: *Capsanthin-capsorubin synthase*; *CAD*: *Cinnamyl alcohol dehydrogenase*; *GLDH*: *L-galactono-1,4-lactone dehydrogenase*. The data points represent the means of three biological replicates ± SD. See Appendix A for primers sequence details.

## Data Availability

Data is available at the Gene Expression Omnibus (GEO) of the NCBI (https://www.ncbi.nlm.nih.gov/geo/, accessed on 5 January 2021). The accession number is GSE165448.

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
