# Peer review of "Genome-Wide Identification and Analysis of the MYB Transcription Factor Gene Family in Chili Pepper (Capsicum spp.)"

_ijms, 2021, doi:10.3390/ijms22052229_

Round 1
Reviewer 1 Report
The manuscript ijms-1084942 is of interest to the readers and could be published prior to some minor revision. Below I comment each section separately:
Introduction: Is well‐written and concise. The hypothesis and purpose of study are clearly and concisely presented. Data are accurate; hypotheses are correctly presented and fully supported by text. Also current references exist that will be of interest to the readers.
Materials and Methods: All research components are present, clearly stated the procedures are clear, concise, and easily replicable; the present manuscript advances knowledge while, all compliance guidelines are met. Tables/figures contribute substantially to content.
Results: Minor rewriting needed to support all items mentioned in Materials and methods there is also some minor repetition of data already included in tables/other text, but rewriting can address this.
Discussion/Conclusion: Study implications and/or limitations are presented but are missing a point(s), which can be included in the conclusion section which requires rewriting and shortening.
Author Response
First of all, we would like to thank Reviewer 1 for the comments to improve our manuscript. Responses to the observations and comments are indicated below.
Results: Minor rewriting needed to support all items mentioned in Materials and methods there is also some minor repetition of data already included in tables/other text, but rewriting can address this.
R.- We want to emphasize that this comment has been taken into consideration and we have addressed this as best we could. Unfortunately, we do not know exactly which parts needed more work terms of the observations and comments made by Reviewer 1, because no marks or indications were made specifically on the text of our manuscript; nevertheless, some changes were introduced in Tables and in the text to try to attend the recommendations. We would be delighted to address any further indication or correction suggested by Reviewer 1.
Discussion/Conclusion: Study implications and/or limitations are presented but are missing a point(s), which can be included in the conclusion section which requires rewriting and shortening.
R.- Some new information was incorporated into the Discussion section, and the Conclusion was shortened in this revised/corrected version of our manuscript.
Reviewer 2 Report
IJMS-1084042
Genome-wide identification and analysis of the MYB transcription factor gene family in chili pepper (Capsicum spp.) by Arce-Rodríguez et al.
The authors performed an extensive genomic analysis to identify genes from the MYB transcription factor family in chili pepper (Capsicum spp.). These analyses include phylogenetic relationships, conserved domain, gene structure organization, motif protein arrangement, chromosome distribution, chemical properties predictions, RNA-seq expression, and RT-qPCR expression assays. The results are very interesting provide new and high quality information for researchers in the field of plant development.
However, two major problems should be solved to improve the manuscript:
Point 1. The author used a double annotation for the genes/proteins in the figures with long and short code numbers (e.g., XP_016574578.1/ CaMYBR1, XP_016577606.1/ CaMYBR13). In particular, Figures 1, 3 and 4 use long codes whereas figures 2, 6 and 7 use short codes. This is especially confusing for readers because in the text short codes are used in all the descriptions.
Considering that all the information on protein codes in contained in the supplementary files, a unique code format should be used in all the figures to avoid confusion.
Point 2. Figures 3 and 4 are too long and lettering too small to be read and therefore, information contained in the figure is not accessible for readers. To improve the accessibility of the information I suggest to replace them in the main text by the supplementary figures S1-1, S1-2, S3, S2-1 and S2-2. These figures contain the relevant information regarding exon-intron structure and motif identification but are focused on Capsicum genes.
Again, gene code format should be unified in all the figures.
Additional questions:
Q1. The authors identified 116 R2R3-MYBs, five 3R-MYBs, 92 MYBs-related, and two atypical MYB 89 (CaMYB5R and CaMYBCDC). However, in the Phylogenetic analysis of the MYB family only 3 out of 92 MYB-related proteins were analysed (text lines 108-109 and figure 1). Please, explained the reasons in the text.
To claim that they performed a complete phylogenetic analysis of the CaMYB family, all MYB-related proteins should be included in the tree. Alternatively, a second tree focused in this group of proteins could be added in a separated panel.
Q2. In a recent paper by Wang et al 2020, published in Front. Genet., 21 December 2020 | https://doi.org/10.3389/fgene.2020.598183, the authors studied R2R3-MYB subfamily genes in Capsicum in relation with capsaicinoid biosynthesis (“Genome-Wide Identification and Capsaicinoid Biosynthesis-Related Expression Analysis of the R2R3-MYB Gene Family in Capsicum annuum L.”). This manuscript should be included in the references and mentioned/discussed in the text.
Q3. In the methods section (lines 568-570) you wrote, “We analyzed the expression of eleven CaMYB genes and five key structural genes of the phenylpropanoid, lignin, capsaicinoid, carotenoid, and vitamin C biosynthesis pathways”. This relevant information on the specific pathways should also be included in the legend of figure 7 and in the results section (2.7. RT-qPCR analysis).
Minor points:
- Please, include in the legend of the figure information regarding species name abbreviation. For instance, At Arabidopsis thaliana, St Solanum tuberosum, Am Anthirrinun majus.
- Figure 5 seems to have a format problem (fruits are not visible) in my version of the manuscript.
- Legend figure 7: please include the tissue where the analysis was made.
- Line 292 in the text contain information of a gene not included in figure 7.
Author Response
We are very grateful for the critical review and the valuable comments and observations made by Reviewer 2 to improve our manuscript. Responses point-by-point are indicated below.
Point 1. The author used a double annotation for the genes/proteins in the figures with long and short code numbers (e.g., XP_016574578.1/ CaMYBR1, XP_016577606.1/ CaMYBR13). In particular, Figures 1, 3 and 4 use long codes whereas figures 2, 6 and 7 use short codes. This is especially confusing for readers because in the text short codes are used in all the descriptions.
Considering that all the information on protein codes in contained in the supplementary files, a unique code format should be used in all the figures to avoid confusion.
R.- We have changed the code format and use short code in all figures in this revised/corrected version of our manuscript.
Point 2. Figures 3 and 4 are too long and lettering too small to be read and therefore, information contained in the figure is not accessible for readers. To improve the accessibility of the information I suggest to replace them in the main text by the supplementary figures S1-1, S1-2, S3, S2-1 and S2-2. These figures contain the relevant information regarding exon-intron structure and motif identification but are focused on Capsicum genes.
Again, gene code format should be unified in all the figures.
R.- We respectfully consider that Figures 3 and 4 represent the core contribution of the analysis of our manuscript and they should be maintained in the main text; therefore, we decided to leave them as they were placed in the former version. To solve the problem we increased the size and sharpness of fonts of Figures 3 and 4, and the readers could magnify them when necessary.
Additionally, the short gene code was applied to all figures in order to unify the format.
Additional questions:
Q1. The authors identified 116 R2R3-MYBs, five 3R-MYBs, 92 MYBs-related, and two atypical MYB 89 (CaMYB5R and CaMYBCDC). However, in the Phylogenetic analysis of the MYB family only 3 out of 92 MYB-related proteins were analysed (text lines 108-109 and figure 1). Please, explained the reasons in the text.
To claim that they performed a complete phylogenetic analysis of the CaMYB family, all MYB-related proteins should be included in the tree. Alternatively, a second tree focused in this group of proteins could be added in a separated panel.
R.- The phylogenetic analysis of the MYB-related subfamily was not included since a quality alignment of all their members is not feasible. Although all MYB-related proteins contained at least one MYB repeat, they were not necessarily similar between subgroups.
Q2. In a recent paper by Wang et al 2020, published in Front. Genet., 21 December 2020 | https://doi.org/10.3389/fgene.2020.598183, the authors studied R2R3-MYB subfamily genes in Capsicum in relation with capsaicinoid biosynthesis (“Genome-Wide Identification and Capsaicinoid Biosynthesis-Related Expression Analysis of the R2R3-MYB Gene Family in Capsicum annuum L.”). This manuscript should be included in the references and mentioned/discussed in the text.
R.- The article of Wang et al. (2020) was cited in the text and the information was incorporated into the Discussion section in this revised/corrected version.
Q3. In the methods section (lines 568-570) you wrote, “We analyzed the expression of eleven CaMYB genes and five key structural genes of the phenylpropanoid, lignin, capsaicinoid, carotenoid, and vitamin C biosynthesis pathways”. This relevant information on the specific pathways should also be included in the legend of figure 7 and in the results section (2.7. RT-qPCR analysis).
R.- The relevant information on the specific pathways was included in the legend of Figure 7 and in the results section 2.7 in this revised/corrected version of the manuscript.
Minor points:
- Please, include in the legend of the figure information regarding species name abbreviation. For instance, At Arabidopsis thaliana, St Solanum tuberosum, Am Anthirrinun majus.
R.- This recommendation was addressed in this revised/corrected version of the manuscript.
- Figure 5 seems to have a format problem (fruits are not visible) in my version of the manuscript.
R.- We have compressed the size of figure 5 without affecting its quality and changed to .jpg format hoping to solve further issues of visualization.
- Legend figure 7: please include the tissue where the analysis was made.
R.- This recommendation was addressed in this revised/corrected version of the manuscript.
- Line 292 in the text contain information of a gene not included in figure 7.
R.- Unfortunately, we could not find this error in the manuscript.